# On the Computational Complexity of Stackelberg Planning and Meta-Operator Verification

**Primary Keywords:** *(4) Theory*

## Abstract

Stackelberg planning is a two-player variant of classical planning, in which one player tries to "sabotage" the other player in achieving its goal. This yields a bi-objective planning problem, which appears to be computationally more challenging than the single-player case. But is this actually true? All investigations so far focused on practical aspects, i.e., algorithms, and applications like cyber-security or very recently for meta-operator verification in classical planning. We close this gap by conducting the first theoretical complexity analysis of Stackelberg planning. We show that in general Stackelberg planning is no harder than classical planning. Under a polynomial plan-length restriction, however, Stackelberg planning is a level higher up in the polynomial complexity hierarchy, suggesting that compilations into classical planning come with an exponential plan-length increase. In attempts to identify tractable fragments exploitable, e.g., for Stackelberg planning heuristic design, we further study its complexity under various planning task restrictions, showing that Stackelberg planning remains intractable where classical planning is not. We finally inspect the complexity of the meta-operator verification, which in particular gives rise to a new interpretation as the dual problem of Stackelberg plan existence.

## Introduction

Stackelberg planning (Speicher et al. 2018a) is a two-player variant of classical planning, where one player (the *leader*) tries to "sabotage" the other player (the *follower*). The leader moves first, committing to an action sequence, which subsequently the follower needs to complete to a plan. The leader's objective is maximizing the follower's optimal plan cost while minimizing her own cost. This type of planning is useful for real-world adversarial settings commonly found in the cyber-security domain (Speicher et al. 2018b; Di Tizio et al. 2023). *Leader-follower search* (Speicher et al. 2018a) is the so far only algorithm paradigm proposed for solving such tasks. In essence, it boils down to a search in the leader state space, solving for every visited leader state the follower's associated classical planning task. Given that exponentially such follower tasks must be solved in the worst case, one might wonder whether Stackelberg planning is in fact computationally more difficult than classical variant. Past work on Stackelberg planning however so far focused on algorithmic improvements rather than studying this question (Torralba et al. 2021; Sauer et al. 2023).

To close this gap, we present the first theoretical investigation of Stackelberg planning's complexity. We show that Stackelberg planning remains PSPACE-complete in general. However, Stackelberg planning with polynomial plan-length bounds is $\Sigma_2^P$-complete, contrasting the NP-completeness of the corresponding classical planning problem (Bylander 1994). Assuming that the polynomial hierarchy does not collapse, this suggests that compilations of Stackelberg planning into classical planning need to come with an exponential increase in plan length.

The analysis of tractable fragments has shown to be an important source for the development of domain-independent heuristic in classical planning (e.g., Hoffmann and Nebel 2001; Domshlak, Hoffmann, and Katz 2015). With the vision of establishing a basis for the development of leader-follower search heuristics, we analyze the complexity of Stackelberg planning under various syntactic restrictions. An overview of our results is given in Tab. 1.

Lastly, we explore a problem related to Stackelberg planning: *meta-operator* (Pham and Torralba 2023) verification. Meta-operators are action-sequence wild cards, which can be instantiated freely for every state satisfying the operator's precondition as long as operator's effects match. Pham and Torralba have cast verifying whether a given action is a valid meta-operator as a Stackelberg planning task. We show that meta-operator verification PSPACE-complete and $\Pi_2^P$-complete under a polynomial plan-length restriction. This gives rise to a new interpretation of the meta-operator verification as the dual problem of Stackelberg planning.

**Note to Reviewers:** This is a short-paper version without proofs. All proofs are in the supplement, which we will publish. Alternatively, if you so desire, we can include all proofs into a long version of paper (see alternative attached).

## Background

**Classical Planning**   We assume STRIPS notation (Fikes and Nilsson 1971). A planning task is a tuple $\Pi = \langle V, A, I, G \rangle$ consisting of a set of propositional *state variables* (or *facts*) $V$, a set of *actions* $A$, an *initial state* $I \subseteq V$, and a *goal* $G \subseteq V$. For $p \in V$, $p$ and $\neg p$ are called *literals*. A *state* $s$ is a subset of $V$, with the interpretation that all state variables not in $s$ do not hold in $s$. Each action $a \in A$ has a *precondition* $pre(a)$, a conjunction of literals, an *add effect* (also called positive effect) $add(a) \subseteq V$, a *delete effect* (neg-

| | Plan existence | | Optimal planning | | METAOPVER |
|---|---|---|---|---|---|
| Syntactic restrictions | PLANSAT | STACKELSAT | PLANMIN | STACKELMIN | |
| $*$ preconds $*$ effects $\lvert\pi\rvert$ not bounded | PSPACE | PSPACE (Theorem 1) | PSPACE | PSPACE (Theorem 2) | PSPACE (Theorem 10) |
| $*$ preconds $*$ effects $\lvert\pi\rvert \in \mathcal{O}(n^k)$ | NP | $\Sigma_2^{\mathrm{P}}$ (Theorem 3) | NP | $\Sigma_2^{\mathrm{P}}$ (Theorem 3) | $\Pi_2^{\mathrm{P}}$ (Theorem 11) |
| 1 precond 1+ effect | NP | $\Sigma_2^{\mathrm{P}}$ (Theorem 4) | NP | $\Sigma_2^{\mathrm{P}}$ (Corollary 1) | – |
| $*+$ preconds 1 effect | P | NP (Theorem 5) | NP | $\Sigma_2^{\mathrm{P}}$ (Theorem 7) | – |
| 0 preconds 2 effects | P | P for $\infty$ effects  (Theorem 6 ) | NP | $\Sigma_2^{\mathrm{P}}$ (Theorem 8) | – |
| 0 preconds 1 effect non-unit cost | P | P for $\infty$ effects  (Theorem 6) | P | NP (Theorem 9) | – |

Table 1: Overview of our complexity results. For comparison, the PLANSAT and PLANMIN columns show the complexity of classical planning under the respective task restrictions, as given by (Bylander 1994). All results prove completeness with respect to the different complexity classes. $*$ means arbitrary number, $+$ only positive, $*+$ arbitrary positive, and $n+$ $n$ positive.

ative effect) $del(a) \subseteq V$, and a non-negative *cost* $c(a) \in \mathbb{N}_0$. $a$ is applicable in a state $s$ iff $s \models pre(a)$. Executing $a$ in $s$ yields the state $s[\![a]\!] = (s \setminus del(a)) \cup add(a)$. These definitions are extended to action sequences $\pi$ in an iterative manner. The cost of $\pi$ is the sum of costs of its actions. $\pi$ is called an $s$-*plan* if $\pi$ is applicable in $s$ and $G \subseteq s[\![\pi]\!]$. $\pi$ is an *optimal* $s$-plan if $c(\pi)$ is minimal among all $s$-plans. An (optimal) plan for $\Pi$ is an (optimal) $I$-plan. If there is no $I$-plan, we say that $\Pi$ is *unsolvable*. Two decision problem formulations of classical planning are considered in the literature. *PLANSAT* is the problem of given a planning task $\Pi$, deciding whether there exists any plan for $\Pi$. *PLANMIN* asks, given in addition a (binary-encoded) cost bound $B$, whether there is a plan $\pi$ for $\Pi$ with cost $c(\pi) \leq B$. Both problems are known to be PSPACE-complete (Bylander 1994).

**Stackelberg Planning**  A Stackelberg planning task (Speicher et al. 2018a) is a tuple $\Pi^{LF} = \langle V, A^L, A^F, I, G^F \rangle$, where the set of actions is partitioned into one for each player. A *leader plan* is an action sequence $\pi^L = \langle a_1^L, \ldots, a_n^L \rangle \in (A^L)^n$ that is applicable in $I$. $\pi^L$ induces the *follower task* $\Pi^F(\pi^L) = \langle V, A^F, I[\![\pi^L]\!], G^F \rangle$. An (optimal) *follower response* to $\pi^L$ is an (optimal) plan for $\Pi^F(\pi^L)$. We denote by $c^F(\pi^L)$ the cost of the optimal follower response to $\pi^L$, defining $c^F(\pi^L) = \infty$ if $\Pi^F(\pi^L)$ is unsolvable. Leader plans are compared via a dominance order between cost pairs where $\langle c_1^L, c_1^F \rangle$ *weakly dominates* $\langle c_2^L, c_2^F \rangle$ ($\langle c_1^L, c_1^F \rangle \sqsubseteq \langle c_2^L, c_2^F \rangle$), if $c_1^L \leq c_2^L$ and $c_1^F \geq c_2^F$. $\langle c_1^L, c_1^F \rangle$ (strictly) *dominates* $\langle c_2^L, c_2^F \rangle$ ($\langle c_1^L, c_1^F \rangle \sqsubset \langle c_2^L, c_2^F \rangle$), if $\langle c_1^L, c_1^F \rangle \sqsubseteq \langle c_2^L, c_2^F \rangle$ and $\langle c_1^L, c_1^F \rangle \neq \langle c_2^L, c_2^F \rangle$. To simplify notation, we write $\pi_1^L \sqsubset \pi_2^L$ if $\langle c(\pi_1^L), c^F(\pi_1^L) \rangle \sqsubset \langle c(\pi_2^L), c^F(\pi_2^L) \rangle$. A leader plan $\pi^L$ is optimal if it is not dominated by any leader plan. Previous works have considered algorithms for computing the set of all optimal solutions, called the *Pareto frontier*.

## Stackelberg Planning Decision Problems

We distinguish between two decision-theoretic formulations of Stackelberg planning, akin to classical planning:

**Definition 1** (STACKELSAT). *Given* $\Pi^{LF}$, *STACKELSAT is the problem of deciding whether there is a leader plan* $\pi^L$ *that makes* $\Pi^F(\pi^L)$ *unsolvable.*

**Definition 2** (STACKELMIN). *Given* $\Pi^{LF}$, *and two binary-encoded numbers* $B^L, B^F \in \mathbb{N}_0$. *STACKELMIN is the problem of deciding whether there is a leader plan* $\pi^L$ *with* $\langle c(\pi^L), c^F(\pi^L) \rangle \sqsubseteq \langle B^L, B^F \rangle$.

Interpreting the leader's objective as rendering the follower's objective infeasible, the first definition directly mirrors the PLANSAT plan-existence decision problem. Similarly, the second definition mirrors PLANMIN in looking for solutions matching a given quantitative cost bound. It is worth mentioning that both decision problems are implicitly looking for only a single point in the Pareto frontier, whereas previous practical works dealt with algorithms computing this frontier entirely. In terms of computational complexity, this difference is however unimportant. In particular, answering even just a single STACKELMIN question does in fact subsume the computation of the entire Pareto frontier – if the answer is no, one necessarily had to compare the given bounds to *every* element in the Pareto frontier.

As in classical planning, STACKELSAT can be easily (with polynomial overhead) reduced to STACKELMIN:

**Proposition 1.** *STACKELSAT is polynomially reducible to STACKELMIN.*

Given that Stackelberg planning is a proper generalization of classical planning, the Stackelberg decision problems are guaranteed to be at least as hard as the respective classical planning decision problem. By applying the same proof idea as the Immerman–Szelepcsényi theorem (Szelepcsényi 1987; Immerman 1988), we can prove that it is also no harder than classical planning in the general case:

**Theorem 1.** *STACKELSAT is PSPACE-complete.*

**Theorem 2.** *STACKELMIN is PSPACE-complete.*

In spite of these results, algorithms for Stackelberg planning are significantly more complicated than their classical planning counterparts. In particular, the results raise the

question of whether it is possible to leverage directly the classical planning methods for solving Stackelberg tasks via compilation. Polynomial compilations necessarily exist as per the theorems, yet, it is interesting to investigate which "side-effects" these might need to have. For example, it is possible any such compilation will have exponentially longer plan, rendering this approach infeasible in practice. In order to investigate these questions, we turn to a more fine granular analysis by considering the complexity under various previously studied syntactic classes of planning tasks.

## Stackelberg Planning under Restrictions

### Polynomial Plan Length

For classical planning, it is commonly known that restricting the length of the plans to be *polynomial* in the size of the planning task description, makes the decision problems become NP-complete.

**Definition 3** (Polynomial Stackelberg Decision). *Given $\Pi^{LF}$ with non-0 action costs, and two binary-encoded numbers $B^L, B^F \in \mathbb{N}_0$ that are bounded by some polynomial $p \in \mathcal{O}(\ell^k)$ for $\ell = |V| + |A^L| + |A^F|$. STACKELPOLY is the problem of deciding whether there is a leader plan $\pi^L$ such that $\langle c(\pi^L), c^F(\pi^L) \rangle \sqsubseteq \langle B^L, B^F \rangle$.*

We restrict the action cost to be strictly positive, ensuring that considering leader and follower plans with polynomial length is sufficient to answer the decision problem. STACKELPOLY is harder than the corresponding classical problem.

**Theorem 3.** *STACKELPOLY is $\Sigma_2^P$-complete.*

This result strongly suggests that a compilation of Stackelberg planning into classical planning is in general not possible without an exponential blow-up of some kind. Namely, suppose it were possible to compile any Stackelberg planning task into classical planning in a way so that the size as well as the length of the plans of the classical planning task can be related polynomially to the size of the Stackelberg task. Suppose the plans of the Stackelberg task are polynomially bounded. Since polynomial length plan existence for classical planning is NP-complete, this would, together with our result, imply that NP $= \Sigma_2^P$, thus collapsing the polynomial hierarchy (Arora and Barak 2007, Theorem 5.6). As this is unlikely given out current knowledge, we hence surmise that such polynomial compilations do not exist. Or in other words: we know that an exponential blow-up in the computation is not avoidable in all circumstances.

### Stackelberg Planning under Bylander's Syntactic Restrictions

Bylander (1994) studied the complexity of classical planning under various syntactic restrictions, drawing a concise borderline between planning's tractability and infeasibility. Bylander distinguishes between different planning task classes based on the number of action preconditions and effects, and the existence of negative preconditions or effects. Table 1 provides an overview of the main classes. Here, we take up his analysis and show that even for the classes where classical planning is tractable, Stackelberg may not be. We consider STACKELSAT and STACKELMIN in this order.

**Definition 4.** *Let $m, n \in \mathbb{N}_0 \cup \{\infty\}$. STACKELSAT$_n^m$ is the problem of deciding STACKELSAT for Stackelberg tasks so that $|pre(a)| \leq m$ and $|add(a)| + |del(a)| \leq n$ hold for all actions $a$. If $m$ is preceded by "+", actions may not have negative preconditions. If $n$ is preceded by "+", actions may not have delete effects. STACKELMIN$_n^m$ is defined similarly.*

We omit $m$ ($n$) if $m = \infty$ ($n = \infty$). We consider only cases where the classical-planning decision problems are in NP. Stackelberg planning is PSPACE-hard when classical planning is.

**Plan Existence**

Bylander (1994) has shown that PLANSAT is already NP-complete for tasks with actions that even have just a single precondition and a single effect. Here we show that the corresponding Stackelberg decision problem is even one step above in the polynomial hierarchy:

**Theorem 4.** *STACKELSAT$_{+1}^1$ is $\Sigma_2^P$-complete.*

Bylander (1994) has shown that PLANSAT is polynomial if only positive preconditions and only a single effect per action are allowed. Even under these restrictive conditions, STACKELSAT however still remains intractable:

**Theorem 5.** *STACKELSAT$_1^+$ is NP-complete.*

Stackelberg plan-existence however becomes easy, when forbidding preconditions throughout. While this class of tasks seems to be trivial at first glance, optimal Stackelberg planning actually remains intractable as we show below.

**Theorem 6.** *STACKELSAT$^0$ is polynomial.*

**Optimal Planning**

As per Proposition 1, optimal planning is in general at least as hard as deciding plan existence. All intractability results shown for STACKELSAT carry over to STACKELMIN. As in all classes analyzed in the previous section, the consideration of polynomially length-bounded plans is sufficient for hardness, $\Sigma_2^P$ yields a sharp upper bound to the complexity of STACKELMIN, as per Theorem 3. In particular:

**Corrolary 1.** *STACKELMIN$_{+1}^1$ is $\Sigma_2^P$-complete.*

The results for STACKELSAT only provide a lower bound to the complexity of STACKELMIN. This lower bound may be strict as demonstrated by Thm. 7 and 8:

**Theorem 7.** *STACKELMIN$_1^{+1}$ is $\Sigma_2^P$-complete.*

**Theorem 8.** *STACKELMIN$_2^0$ is $\Sigma_2^P$-complete.*

Optimal Stackelberg planning remains intractable even when all actions have no preconditions and may have only at most one effect.

**Theorem 9.** *STACKELMIN$_1^0$ is NP-complete in general, but polynomial when additionally assuming unit cost.*

## Complexity of Meta Operator Verification

Pham and Torralba (2023) have recently leveraged Stackelberg planning for synthesizing *meta-operators* in classical planning. Meta-operators, like macro-actions (Fikes and Nilsson 1971), are artificial actions that aggregate the effect of action sequences, therewith introducing shortcuts in state-space search. Formally, we are given a classical planning

task $\Pi$ and an action $\sigma$ that is not in $\Pi$'s action set. $\sigma$ is a *meta-operator for $\Pi$* if, for every state $s \models pre(\sigma)$ that is reachable from $I$, there exists a sequence $\pi$ of $\Pi$'s actions such that $s[\![\sigma]\!] = s[\![\pi]\!]$. Whether a given $\sigma$ is a meta-operator can be *verified* by solving a Stackelberg planning task.

Here, we consider the question whether using an expressive and computationally difficult formalism like Stackelberg planning is actually necessary. For this, we determine the computational complexity of meta-operator synthesis and compare it to that of Stackelberg planning, and based on this analysis point out an interesting connection.

**Definition 5** (Meta-Operator Verification). *Given $\Pi$ and a fresh action $\sigma$. METAOPVER is the problem of deciding whether $\sigma$ is a meta-operator for $\Pi$.*

Like for Stackelberg planning, the complexity of meta-operator verification in general remains the same as that of classical planning:

**Theorem 10.** *METAOPVER is PSPACE-complete.*

In other words, meta-operator verification could as well be compiled directly into a classical rather than a Stackelberg planning task. But how difficult or effective would such a compilation be? To shed light on this question, we again turn to a length bounded version of the problem.

**Definition 6** (Polynomial Meta-Operator Verification). *Given $\Pi$ with non-$0$ action costs, a fresh action $\sigma$, and two binary-encoded numbers $B^P, B^M \in \mathbb{N}_0$ that are bounded by some polynomial $p \in \mathcal{O}(\ell^k)$ for $\ell = |V| + |A|$. polyMETAOPVER is the problem of deciding whether for all states $s \models pre(\sigma)$ reachable from $I$ with a cost of at most $B^P$, there exists $\pi$ with $c(\pi) \le B^M$ and $s[\![\pi]\!] = s[\![\sigma]\!]$.*

The parameters $B^P$ and $B^M$ define the perimeter around the initial state respectively the reached state under which the meta-operator conditions are to be verified. As for Stackelberg planning, we require that the cost of all actions is strictly positive, which together with the cost bounds ensures that the radius of the perimeter is polynomially bounded.

Polynomial meta-operator verification too is on the second level of the polynomial hierarchy. We again point out that, under the assumption that the polynomial hierarchy does not collapse, this result shows that all classical-planning encodings of meta-operator verification generally need to come with an exponential explosion of some kind.

**Theorem 11.** *polyMETAOPVER is $\Pi_2^P$-complete.*

Note that polyMETAOPVER is therefore in the co-complexity-class of polynomial Stackelberg plan-existence, i.e., they belong to co-classes on the same level of the polynomial hierarchy. This may not be surprising given that meta-operator verification can indeed be seen as the dual of Stackelberg plan existence: while the latter asks for the existence of a (leader) action sequence where all induced (follower) action sequences satisfy some property, meta-operator verification swaps the quantifiers.

We want to point out that the duality between METAOPVER and STACKELSAT can be exploited further, showing analogous results for Bylander's (1994) task classes. Contrary to Stackelberg planning, however, the identification of tractable fragments is less useful for meta-operator verification due to the lack of the monotonicity invariance of the meta-operator condition. An action being a meta-operator in a task abstraction does not imply that the action is a meta-operator in the original task, and vice versa. We hence do not further explore this analysis here.

## Related Work

Stackelberg planning is related to conformant and conditional planning, extensions of classical planning by state and/or action outcome uncertainty. Under the restriction to deterministic actions, both can be seen as a special case of Stackelberg planning using the leader-reachable states as an encoding of the initial belief. With this interpretation, STACKELSAT is false iff the conditional planning task is solvable. If the follower is restricted to use the same plan independent of the leader actions, we would have a model for conformant planning.

In the general case, conditional planning under partial observability and with conditional effects is EXPSPACE complete (Rintanen 2004). Both conformant and conditional planning have been investigated under the restriction to only polynomially long plans, like we did here. Rintanen (1999) showed that polynomially-length-bounded conditional STRIPS planning $\Pi_2^P$ complete, the co-result to our Thm. 3. His hardness proof uses a similar proof idea as ours, with technical differences owed to the different planning formalism. Bonet (2010) studied conditional planning with non-deterministic actions, proving that polynomially bounded plan existence for conditional plans with at most $k$ branching points is $\Sigma_{2k+k}^P$-complete. Stackelberg planning corresponds $k = 1$, the difference between determinism and non-determinism causing the $\Sigma_2^P$ vs. $\Sigma_4^P$ complexity results.

For conformant planning, Baral, Kreinovich, and Trejo (2000) showed that plan existence is $\Sigma_P^2$-complete, if conditional effects are allowed. Turner (2002) considered conditional and conformant planning, but his formalism supported arbitrary boolean formulae as conditions, making length-1 plan existence already NP-complete.

No prior work on conformant/conditional planning considered any of Bylander's syntactical restrictions. Further, Stackelberg planning differs from conditional/conformant planning in using a more complex compact description of the "relevant" states through reachability.

## Conclusion

Stackelberg planning remains PSPACE-complete like classical planning in general, but is $\Sigma_2^P$ complete under a polynomial plan-length bound. Hence, unless the polynomial hierarchy collapses at its first level, it is not possible to compile Stackelberg planning into classical planning without exponential blow-up. We showed that Stackelberg planning remains intractable even under various syntactical restrictions. Lastly, we have proven similar results for meta-operator verification, showing that it is PSPACE-complete in general and $\Pi_2^P$-complete for the polynomial plan-length bounded case, implying the same type of results for it.

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
