# OpenReview forum: "On the Computational Complexity of Stackelberg Planning and Meta-Operator Verification"
_icaps-conference.org/ICAPS/2024/Conference — ICAPS 2024_

### Official Review · Reviewer_wCuQ · 2024-01-10

**Significance And Importance:** 2
**Soundness:** 4
**Novelty:** 2
**Clarity:** 3
**Overall Evaluation:** 1
**Confidence:** 5

**Weaknesses:**

0: Minor weaknesses requiring some work to be addressed for the paper to be accepted.

**Contributions Of The Paper:**

The paper proves some computational complexity results for Stackelberg planning -- a flavor of planning where a leader executes a sequence of actions, and then a follower tries to find a plan from there. In general, Stackelberg planning is as hard as classical propositional planning. The authors also introduce some more fine-grained complexity results (based on structural restrictions of actions) and show that some of these restrictions lead to problems within the second level of the polynomial hierarchy. The authors also study the related problem of meta-operator verification (given a "meta-operator" O, for every state s where O is applicable, could s[O] be reached by another sequence of "non-meta" operators?) and show that it is also hard and some of the more restricted cases also fall into the polynomial hierarchy.

**Ethical Considerations:**

(1) Not Applicable: The paper does not have any ethical considerations to address

**Nomination For Best Paper:**

No

**Questions For Authors:**

I have no questions.

**Reproducibility:**

0: N/A - nothing to reproduce.

**Strengths Of The Paper:**

- The paper addresses the specific problem of Stackelberg planning from a theoretical perspective, which has not been done before.
- The paper is mostly clear and well-written.
- To the best of my knowledge, the proofs and results are sound.

**Weaknesses Of The Paper:**

I have one larger issue with the paper (novelty and connection to related work) and some more technical problems. The main problem is that Stackelberg planning is naturally seen as a simplification of FOND planning (almost any two-player game on graphs could be viewed as such) but this is never even acknowledged. Overall, I still recommend acceptance as a short paper, because it can be useful to have a concrete reference for the proofs and theorems, but I strongly recommend a revision to make it clearer that several of these results/proofs are related to previously studied and well-established concepts in planning.

In more detail:

Several of the proofs and results are adaptations/simplifications of canonical proofs in planning and theoretical computer science. This is OK, but this could be acknowledged more clearly. For example, alternating Turing Machines and representation of QBF are used over and over in planning results (some of which you cited), and making this connection would make the results much more straightforward, particularly in a dense short paper where any hint will help the reader. Moreover, Stackelbeg planning is just a simplified case of two-player games, but you never put it in this light. You always mention it as a somehow "practically harder" version of classical planning, but it is much more natural to think of it as a simplification of probabilistic/non-deterministic or adversarial planning -- which are never mentioned. In particular, I think the relation to non-deterministic planning *must* be discussed in the paper. Putting it forcefully, one can think of Stackelberg planning as a very simplified version of FOND planning.

Talking about non-deterministic planning, the seminal paper by Littman [1997] gives one more example of the relation to previous work: Thm 1 and Thm 2 of his paper present ideas that are very similar to your proofs. The proofs can also be modified to show that Stackelberg planning is a simpler case of non-deterministic/probabilistic planning. We reduce Stackelberg planning to non-deterministic planning as follows: The new actions will be the leader actions plus two extra actions CHANGE-TURN and FOLLOWERS-ACTION. Add a propositional variable LEADERS-TURN to the precondition of all actions that were originally from the leader (and make LEADERS-TURN true in the initial state). Action CHANGE-TURN has precondition LEADERS-TURN, and effect (not LEADERS-TURN and FOLLOWERS-TURN). Action FOLLOWERS-ACTION has precondition (FOLLOWERS-TURNS and not TERMINATED), and a non-deterministic effect (oneof (not FOLLOWERS-TURN), E_1, E_2, ..., E_n) where E_i is is the conditional effect (when Pre_i Eff_i) where Pre_i and Eff_i are the precondition and effect of the i-th follower action. The goal is ((not LEADERS-TURN) and (not FOLLOWERS-TURN) and (not GOAL)), where GOAL is the original goal of the follower. The original Stackelberg task is only satisfiable if the non-deterministic task reaches the new goal with certainty. Note that the new goal represents that it is not the turn of any of the players (so in some sense they both "passed") but the original goal is still not reached. You can use the same idea for probabilistic planning using arbitrary non-zero probabilities for the effects.

A technical note: The proof of Theorem 1 must be rewritten. You mention using Immerman-Szelepcsenyi but it needs more details, and I believe you mixed the Immerman-Szelepcsenyi theorem with the Savitch theorem. The key here is that your algorithm guesses a plan for the leader, and then checks that no follower plan exists. In other words, the second step checks *unsolvability* instead of solvability. Alone, Savitch's algorithm in the second part (follower's plan) will give you nothing, because it can answer YES in poly-space (i.e., a plan exists), not NO (i.e., task is unsolvable). But here is where Immerman-Szelepcsenyi comes into play -- which shows that coPSPACE = PSPACE -- and you can indeed decide unsolvability in PSPACE.

Last, I was a bit confused by the discussion about meta-operators. The connection to Stackelberg planning is clear, but one key difference is that in METAOPVER \sigma is given. There is nothing analogous given to STACKELSAT. This difference should also be pointed out so it just not sound like it is direct to complement one for the other. Right now it reads as if METAOPVER is the complement of STACKELSAT, as UNSAT is the complement of SAT, for example.

Minor:
In the paper:
- You frequently use the word "compilation" instead of "reduction". I think in almost every case you want to say "reduction", which assumes that the solution sizes are polynomially preserved, while "compilation" does not.
- Starting from Thm 4, there is a mismatch in the numbering between the paper and the appendix.
- "As per Proposition 1, optimal planning is in general at least as hard as deciding plan existence" -> say "optimal *Stackelberg* planning", as this is not the case in general for other flavors of planning (e.g., deterministic planning using schematic actions and other fragments for which plan existence in EXPSPACE-complete or beyond)
-  "STACKELSAT is false" -> this should be rewritten
- "\Sigma^2_P" -> "\Sigma^P_2"
- Not sure what you mean by the last sentence of the related work section.

In the appendix:
- "alternating Turing Machine" -> "alternating *polynomial-time* Turing Machine"
- Theorem 4 does not exist.
- "Corrolary" -> "Corollary"
- In the proof of Thm 12, the notation for s^P[[\sigma]] used the wrong latex symbol.

---

> ### Author Rebuttal · Authors · 2024-01-28
>
> We thank the reviewer for the very detailed comments and the reduction proof. We acknowledge that our statements regarding the connection of Stackelberg planning to other planning problems types are not ideal, as also pointed out by the other reviewers. We will revise the related work section so as the abstract and introduction to make absolutely clear the relation to FOND/contingent planning and general two-player games.
>
> Thanks for the additional reference. We want to note, though, that while Littman's results mirror ours (EXPTIME and PSPACE for the general and poly-bounded case respectively), his proof-techniques are different: for his Theorem 1, he uses a reduction from the G4 game and a reduction from QBF for his Theorem 2. We will add this to our related work discussion.
>
> Regarding proof of Thm 1, relation between METAOPVER and STACKELSAT, and the typos: thanks for pointing them out. We will fix this in the final version.

---

### Official Review · Reviewer_fXm1 · 2024-01-22

**Significance And Importance:** 2
**Soundness:** 3
**Novelty:** 3
**Clarity:** 3
**Overall Evaluation:** 2
**Confidence:** 3

**Weaknesses:**

2: No major or minor weaknesses.

**Contributions Of The Paper:**

The paper theoretically analyzes the complexity of the Stackelberg Planning Task. It is a two-player adversarial task where the leader tries to thwart the follower’s goal. Previous works have mainly dealt with algorithmic enhancements and applications of the model, leaving the complexity question open until this paper.

The paper investigates two problems, STACKELSAT and STACKLEMIN, and proves that they are both PSPACE-complete. The first problem is whether there is a leader plan that can stop the follower from reaching the goal, and the second asks for a leader plan that dominates the specified cost bounds. All the proof sketches are included in the supplementary material.

The paper also demonstrates the complexity results for specific specializations, like polynomial plan length restriction, and the ones that Bylander investigated in his seminal paper on the complexity of STRIPS Planning. Finally, the paper explores the task of detecting meta-operators for classical planning, for which Stackelberg Planning has been used previously. It discusses the computational complexity of meta-operator verification and the duality of meta-operator verification and Stackelberg planning.

**Ethical Considerations:**

(5) Excellent: The paper comprehensively addresses all of the applicable ethical considerations

**Nomination For Best Paper:**

No

**Questions For Authors:**

-

**Reproducibility:**

0: N/A - nothing to reproduce.

**Strengths Of The Paper:**

The paper presents a theoretical analysis of the complexity of Stackelberg Planning, which, to my knowledge, is an original and important contribution.

Stackelberg Planning arises in cybersecurity and other adversarial problems that can be encoded as a Stackelberg planning task. The results would be significant for researchers working in the domain.

The paper is well-structured. The contributions are well-explained and straightforward to understand.

**Weaknesses Of The Paper:**

Since the paper's primary contribution is theorems describing the theoretical complexity results, I suggest that the proofs of essential theorems be included in the main paper.

---

> ### Author Rebuttal · Authors · 2024-01-28
>
> We thank the reviewer for the positive feedback.
>
> We agree that it would be desirable to include proofs in the main paper. We however believe that the paper is already written very densely. When sticking to the short paper format, additional details could only be added if one excludes some of our results. We find this even less desirable. But, if the reviewers point out parts that can be omitted in place of including at least some proofs, or if the reviewers agree that this work should rather be published as a long paper, including all proofs, (cf. our comment in the letter accompanying the submission), we will happily make the necessary changes for the final version.

---

### Official Review · Reviewer_BbYp · 2024-01-22

**Significance And Importance:** 2
**Soundness:** 3
**Novelty:** 1
**Clarity:** 3
**Confidence:** 4

**Weaknesses:**

-1: Major weaknesses requiring significant work to be addressed for the paper to be accepted.

**Contributions Of The Paper:**

The paper analyzes the main decision problems of "Stackelberg planning" (which is adversarial planning with two players, with one sequence of moves for each), including the general decision problem (ground actions, unlimited plan lengths), same with polynomial plan length, and syntactic special cases following Bylander 1994.

**Ethical Considerations:**

(5) Excellent: The paper comprehensively addresses all of the applicable ethical considerations

**Nomination For Best Paper:**

No

**Overall Evaluation:**

-2: (reject)

**Questions For Authors:**

Q1 Why do you say that Stackelberg planning is a "variant of classical planning"? Wouldn't it be far more accurate to say that it is a special case of contingent and game-theoretic planning problems? In adversarial 2-agent planning there are two agents that alternate with their moves. In your problem one agent makes multiple moves, and then the other agent makes multiple moves.

Q2 What do you mean by "Stackelberg planning differs from conditional/conformant planning in using a more complex compact description of the 'relevant' states through reachability"?

**Reproducibility:**

4: Authors promise to release code and domains (whichever apply).

**Strengths Of The Paper:**

The results have not been published before. There may be important applications for some of the results.

The syntactic restrictions by Bylander 1994 have not been earlier analyzed for contingent planning. (However, they don't seem to be very interesting, as much of contingent planning involves non-deterministic actions, and even in classical planning they have had few applications, reducing their importance.)

**Weaknesses Of The Paper:**

There are two main weaknesses: the novelty is limited due to the similarity to earlier results on contingent and adversarial planning, and the results are not clearly positioned w.r.t. earlier works. Specifically, Stackelberg planning is included in non-deterministic contingent planning, with a simple reduction. One could also say that as a narrow special case of contingent and adversarial planning this subclass is somewhat unusual and not very well motivated. A general weakness would of course be that "Stackelberg planning" as a concept seems a bit superfluous and unnecessary, because "adversarial planner" already fully covers it. Presumably many techniques useful for the former could benefit also the latter, at least if suitably generalized.

The paper should position "Stackelberg planning" in the bigger picture of non-deterministic and adversarial planning already in the abstract and introduction. Saying that it is a "variant of classical planning" is misleading, as classical planning is by definition single-agent, and, instead, Stackelberg is a special case of general (non-deterministic) contingent planning and adversarial planning problems. (Any form of planning could similarly and misleadingly be characterized as "a variant of classical planning"!)

Both alternating Turing machines and Turing machines with oracles are closely related to 2-player games, and the two main results, the PSPACE-completeness of the unlimited Stackelberg game and the SigmaP2-completeness when the game plays are polynomially long, and immediate through PSPACE=PSPACE^PSPACE and SigmaP2=NP^NP. One could trivially generalize these to three rounds, with the first and the third round with the same player, and obtain problems complete for PSPACE=PSPACE^PSPACE^PSPACE and NP^NP^NP=SigmaP3. And so on.

The proof techniques are in many cases similar/same as some of the earlier work, which reduces the novelty value.

There is a simple reduction from Stackelberg planning to contingent planning: Take the leader actions as is but with an additional precondition LEADER (these are all deterministic actions, so classical PDDL) and include an additional action that cedes control to the follower by setting LEADER false. The follower actions are represented by a single non-deterministic action, with the precondition (and (NOT LEADER) (NOT TERMINATE)), and effect (ONEOF e1 e2 ... en (TERMINATE)), where each ei is of the form (when P E) where P is the precondition and E is the effect of one of the follower actions. Here TERMINATE is an auxiliary variable that is initially FALSE. LEADER is initially TRUE.
So first a sequence of leader actions are taken, then control is ceded to the follower, and after that the single action for the follower is iterated arbitrarily many times.
The goal formula is (and TERMINATE (NOT G)), where G is the original goal.
Now an instance is in STACKELSAT if and only if the non-deterministic planning problem has a plan that reaches goals with certainty (with a "strong cyclic" plan): the leader can take any action sequence, and after that the non-deterministic action is iterated one or more times, and no matter when termination gets non-deterministically chosen, one has to be in a state that falsifies G.

The paper is quite technical, and although possible applications are hinted at, in the paper there are none. The paper would be far more interesting if the results were used in a convincing application that was thoroughly fleshed out.

On line 52, SigmaP2-completeness only implies that there is no guaranteed polynomial mapping from your problem with polynomial length restrictions to classical planning with polynomial length restrictions. Even if such mappings cannot always be polynomial, there are certainly cases where polynomiality is easily achieved. One example is when the follower can only take an action sequence with a constant bound, so that you could then reduce the follower to a new goal, and then use classical planning as is. Even without a constant bound on the follower's plans, backward chaining could under some syntactic or other restrictions generate a simple polynomial size goal.

In lines 331-335 it is claimed that Conformant and Contingent planning are *special* *cases* of Stackelberg planning if actions are deterministic. First, conditional planning has never been seriously considered with full observability and deterministic actions only, so there is hardly any prior works that would be so restricted that they could be embedded in Stackelberg planning. One either has non-deterministic actions, or partial observability. Second, conformant planning even with deterministic actions is far harder than Stackelberg planning, and hence the innocent looking condition ""the follower is restricted to use the same plan independent of the leader actions" violates the basic assumptions of Stackelberg planning, and is hence highly misleading, as it in no way makes conformant planning a special case of Stackelberg planning.
In fact, Stackelberg planning can be easily embedded in FOND planning (contingent planning with full observability and non-deterministic actions), as pointed out earlier, and this embedding works both for STACKELSAT and STACKELOPT, contrarary to what the author claims in the rebuttal.

Please correct Sigma2P to SigmaP2 on line 357.

Line 360: Arbitrary Boolean formulas as preconditions does not change the complexity of contingent, conformant or classical planning. Please clarify what you refer to with "conditions".

---

> ### Author Rebuttal · Authors · 2024-01-28
>
> We thank the reviewer for the detailed comments. We apologize for not having made more clear the relation of Stackelberg planning and FOND/contingent planning respectively two-player games in general in the related work section. We will revise that section so as the abstract and introduction to emphasize this point.
>
> General comments:
>
> We do not fully agree with the reviewer on
>
> Novelty: While it is true that some proof ideas are similar to complexity proofs in prior work (as we acknowledge in the related work section), they however have not been connected to Stackelberg planning; in fact, a theoretical analysis of this planning variant is non-existing. Moreover, there are multiple results (e.g. Thm 4-11) that do not appear in this form in previous works.
>
> Motivation: Stackelberg planning is a very well motivated "special case of FOND/contingent planning" as evident from various applications, which also makes its theoretical analysis very much relevant. We also want to point out that the relation of Stackelberg planning and FOND/contingent planning is limited to STACKELSAT; while from a practical usage perspective STACKELOPT is the more interesting problem and, as we show in the paper, there can be differences between the two.
>
> Trivial proof: We do not see why the PSPACE/SigmaP2 complexity results should follow trivially from the observation that PSPACE^PSPACE=PSPACE and SigmaP2=NP^NP. This may yield the basis for an alternative proof, but it does not constitute the whole proof argument.
>
> Reduction
>
> Thanks for the sketch. We want to remark that this only shows that STACKELSAT is a subclass of contingent planning, yielding an upper bound on the complexity but no lower bound, which are the main results of our paper. We will make the connection clear.
>
> Questions:
>
> Q1 Stackelberg planning lying in between classical planning and FOND/game-theoretic planning, one can certainly view it both as a generalization of the former and as a specialization of the latter. We will revise the text accordingly.
>
> Q2 In conditional/conformant (c/c) planning, the initial state is only partially observable. This corresponds in Stackelberg planning to the possible initial states of the follower task. In c/c planning, the initial belief is typically defined succinctly as a Cartesian combination of state features. In Stackelberg planning, the set of possible follower initial states are the leader-reachable states, for which in general no such compact representation exists.

---

### Meta-Review · Area_Chair_sMzP · 2024-02-06

**Recommendation:** Accept (Poster)
**Confidence:** 3

**Metareview:**

The paper analyzes the theoretical complexity of Stackelberg planning. Since Stackelberg planning is a limited special case of FOND planning, the scope of the contribution is somewhat narrow (mostly interesting for those working on this fragment). The ideas behind the complexity proofs are mostly known from the more limited/more general formalism. This means that the work is not too original but it could also be considered as positive, because for many readers it makes it very easy to follow the proofs (it's definitively better than coming up with some obscure new proof idea just for the sake of originality). The reviewers are happy with the clarity of the paper but some of them oppose the presentation of Stackelberg planning as generalization of classical planning (as was done in earlier publications on the topic). Instead it should be viewed as a special case of FOND planning and general adversarial planning problems. The authors promise that they will change the presentation accordingly in the camera-ready version but from the rebuttal, the corresponding reviewers are not convinced that they will be able to do this properly.

Pros:
- clarifies the previously unknown theoretical complexity of Stackelberg planning
- sound and clearly written

Cons:
- limited scope
- relationship to FOND and general adversarial planning not properly discussed
- Results are not surprising and the proofs are similar to the ones from the related formalisms.
- Reviewers see opportunities for more direct/elegant proofs (exploiting the relation to the more special and more general formalisms).
- In one proof the authors refer to the Immerman-Szelepcsenyi theorem but none of the reviewers sees how this is relevant there.

**Ethical Considerations:**

(1) Not Applicable: The paper does not have any ethical considerations to address